# Using the IPcase Index with Inflection Points and the Corresponding Case Numbers to Identify the Impact Hit by COVID-19 in China: An Observation Study

**DOI:** 10.3390/ijerph18041994

**Published:** 2021-02-18

**Authors:** Lin-Yen Wang, Tsair-Wei Chien, Willy Chou

**Affiliations:** 1Department of Pediatrics, Chi-Mei Medical Center, Tainan 700, Taiwan; yen3546@yahoo.com.tw; 2Department of Childhood Education and Nursery, Chia Nan University of Pharmacy and Science, Tainan 700, Taiwan; 3School of Medicine, College of Medicine, Kaohsiung Medical University, Kaohsiung 800, Taiwan; 4Department of Medical Research, Chi-Mei Medical Center, Tainan 700, Taiwan; smile@mail.chimei.org.tw; 5Department of Physical Medicine and Rehabilitation, Chi Mei Hospital Chiali, Tainan 700, Taiwan

**Keywords:** item response theory, ogive curve, absolute advantage coefficient, infection point, COVID-19, forest plot, Kano diagram, choropleth map

## Abstract

Coronavirus disease 2019 (COVID-19) occurred in Wuhan and rapidly spread around the world. Assessing the impact of COVID-19 is the first and foremost concern. The inflection point (IP) and the corresponding cumulative number of infected cases (CNICs) are the two viewpoints that should be jointly considered to differentiate the impact of struggling to fight against COVID-19 (SACOVID). The CNIC data were downloaded from the GitHub website on 23 November 2020. The item response theory model (IRT) was proposed to draw the ogive curve for every province/metropolitan city/area in China. The ipcase-index was determined by multiplying the IP days with the corresponding CNICs. The IRT model was parameterized, and the IP days were determined using the absolute advantage coefficient (AAC). The difference in SACOVID was compared using a forest plot. In the observation study, the top three regions hit severely by COVID-19 were Hong Kong, Shanghai, and Hubei, with IPcase indices of 1744, 723, and 698, respectively, and the top three areas with the most aberrant patterns were Yunnan, Sichuan, and Tianjin, with IP days of 5, 51, and 119, respectively. The difference in IP days was determined (χ2 = 5065666, df = 32, *p* < 0.001) among areas in China. The IRT model with the AAC is recommended to determine the IP days during the COVID-19 pandemic.

## 1. Introduction

Coronavirus disease 2019 (COVID-19) occurred in Wuhan in December 2019 [1] and rapidly spread around the world. Which countries/regions were severely affected by COVID-19 is one of the most frequently asked questions. Using the number of reported cases is common, but not fair [2,3]. Several pandemic-prevention measures such as lockdowns, quarantines, mask-wearing, social distancing, contact reduction, and triggering bold policies on containment and mitigation have been implemented in several countries to flatten the curve of COVID-19 [4] cases and to decrease the strain on the public and health care systems in their areas as much as possible. As such, to what extent the effective control of COVID-19 mitigates the strain during the outbreak is worthy of evaluation.

An index to measure the attainment of the effective control of COVID-19 is thus required for development; similar to that is the use of bibliometric indicators (e.g., h- and x-index [5,6]) that consider both publications and citations for the evaluation of individual research achievements [7,8]. This means that the impact of COVID-19 can be measured from the two perspectives of inflection point (IP) days and the corresponding cumulative numbers of infected cases (CNICs), excluding those excessive CNICs and days of struggling to fight against COVID-19 (SACOVID).

The SACOVID variable can be measured by the core area under the CNICs until the IP days. The IP discussed in previous studies [9,10,11,12,13] verified the effective control of COVID-19. However, the applications and determination of the IP days on SACOVID were not described in detail (such as providing an MP4 video and the original data to illustrate the approach in a study). South Korea, for example, is one of the few countries in the world to have strongly maintained a flat infection curve for more than 50 days, but no such approach was proposed to determine the IP when extended to a further 50 days in the previous study [13].

IP refers to a point on a smooth plane curve where curvature changes sign from an increasing concave (concave downward) to a decreasing convex (concave upward) shape, or vice versa [14]. The CNIC in a country/region can be modeled on an ogive curve (OC) to illustrate the pattern and forecast a future epidemic. The IP is defined at the moment of the outbreak to decrease after a peak [15]. However, there was still no agreement in determining the IP days, until now.

Several researchers [16,17,18,19,20,21,22,23] have proposed using mathematical models to predict the number of COVID-19 cases, and the others investigated the IP [9,10,11,12,13]. None of the researchers properly used the IP days to compare the SACOVID capabilities in the effective control of COVID-19 because of the difficulties in evaluating IP days [10]. The mean number of cases on different days yields significantly different IP days, even though the daily number of confirmed cases (e.g., in the previous 2 or 7 days) can be applied to estimate the provisional IP days [12,24]. Developing a mathematical model to determine the IP days during the COVID-19 pandemic is important.

Building a predictive model, determining the IP days, and visualizing the modeling process for each country/region are the challenges that we encountered. Although many mathematical models have been prosed [24], all of these emphasized a model accurate to the epidemic outbreak instead of diagnosis tools (e.g., examining the most unexpected pattern to the model). The item response theory (IRT) [25,26] is another mathematical probability model with an OC to explain the relationship between items (an epidemic in this study) and the person (a country in this study) using two parameters (e.g., slope and location on an OC), different from those modeling approaches in previous studies [16,17,18,19,20,21,22,23] using numerous parameters in their model without the property of diagnosis to select the most best-fit and misfit regions to the model. 

Each country (or region) responds to the situation during the COVID-19 pandemic, which is similar to that in which one person answers one question (or item) in a test (or questionnaire) based on a statistical probability theory. The item characteristic curve (ICC or the OC) is the trajectory route of CNIC (on the *y*-axis) along with ability measures (on the *x*-axis as the inflected days extended). The infection days can be converted into a unit (named probit or logit [27,28]) between −5 and 5 on a continuum scale (on the *x*-axis from the left to the right side). The IP can then be determined at the moment of the outbreak to decrease around the IP [15]. The concept of importance is to create an OC for each country/area using an objective approach (e.g., modeling parameters) to evaluate the IP.

This study aimed (1) to apply the IRT model to determine the IP days and the corresponding CNICs, (2) to develop an index that can be used to evaluate the impact of COVID-19, and (3) to compare the differences in SACOVID across countries/regions in the world and provinces/metropolitan cities/areas in China. 

## 2. Materials and Methods

### 2.1. Data Source

The COVID-19 CNICs were downloaded from the GitHub website [28] for 33 provinces/metropolitan cities/areas in China on we November 2020 (see Appendix A). All downloaded data are publicly released on the website [29]. Ethical approval was not necessary for this study because all the data were obtained from the GitHub website.

### 2.2. Overall Concepton IRT and IP on an Ogive Curve

#### 2.2.1. The IRT Probability Model

Figure 1 shows the IRT model where the OC was drawn together with the standardized infected days (denoted by θ) on the *x*-axis for a region using Equation (1). The constant factor (1.7) is an adjustment from a probit (Z-score) to a logit scale [27,28]. The corresponding probability (P(θ)) is shown at the left on the *y*-axis.

Parameters a and b represent the discrimination (i.e., the slope), the item (the epidemic situation for a region), and difficulty (i.e., a location toward the left means the outbreak occurred at an earlier stage and to the right indicates the outbreak extended to a later stage). These two parameters were set between 0 and 4 (i.e., the slope) and −5 and 5, respectively, when modeling the epidemic situation for each area. Readers are advised to manipulate these two parameters on their own at the link in [30].
(1)$$P\left(\theta \right)=\frac{1}{\;1\;+{\;e}^{-1.7*a\left(\theta -b\right)}}=\frac{e^{1.7*a\left(\theta -b\right)}}{1\;+{\;e}^{17*a.\left(\theta -b\right)}},\;$$

#### 2.2.2. CNICs Transformed to a Percentage

The CNICs were transformed into an observed percentage(OPi) from 0 to 1 [31,32,33] shown at the right on the *y*-axis (see Equation (2) and Figure 1), where Oi denotes the observed CNIC and the maximum and minimum are denoted by Max and Min, respectively.
(2)$$\mathrm{OPi}=\frac{\left(\mathrm{Oi}\;-\;\mathrm{Min}\right)}{\left(\mathrm{Max}\;-\;\mathrm{Min}\right)},$$

The parameter θ in Equation (1) was transformed from the infected days to the control ability of COVID-19 on the *x*-axis from −5 to 5 using Equation (3).
(3)$$\theta =-5+\left(n_{i-1}\right)\times \frac{\left(5-\left(-5\right)\right)}{N},\;$$
where *N* is the observed days and ni represents the ith day. The probability (denoted by the expected percentage (Ep)) in Equation (1) can be obtained if parameters a and b are known. The a and b model parameters will be defined in Section 2.3.

#### 2.2.3. The Feature of an Ogive Curve

The IP is located between Stages II (outbreak from Points O to A) and III (post-peak from Points B to Q) [15]. Figure 1 shows the relationship between IP days and the CNICs present on the *x*-axis and *y*-axis on the right, respectively.

#### 2.2.4. Transforming EPi Back to the Original CNIC

On the basis of Equation (2), the expected CNIC can be obtained using Equation (4):(4)Expected CNICi=EPi×(Max − Min)+Min, 
where EPi is denoted by P (θ) in Equation (1).

### 2.3. Building the Model and Estimating Parameters

Several formulas and functions (on the basis of Equations (1)–(4)) were put in Microsoft Excel and demonstrated in MP4 videos (see Appendix A and Appendix B).

#### 2.3.1. Properties of the Ogive Curve

The earlier IP drives the OC toward the left. The IP-day variable thus affects the SACOVID (Figure 1) [30].

#### 2.3.2. Parameter Estimation

The Microsoft Solver add-in tool was used to estimate the parameters (Appendix C) through the following steps:A.Objective:To minimize the total residuals using the Microsoft function below:(5)$$\mathrm{SUMXMY}2\;([\mathrm{OPi}\;-\;\mathrm{Epi}]\;\times \;[\mathrm{OPi}\;-\;\mathrm{Epi}])\;=\;\overset{n}{\underset{i=1}{\sum }}{\left(\mathrm{OPi}–\;\mathrm{Epi}\right)}^{2}.$$B.Parameters to estimation:Parameters a and b in Equation (1) are calibrated in the model.C.Constrained terms:The parameters (i.e., a and b) are set in a range between (0, 4) and (−5, 5), respectively.D.Data arrangement:The observed CNICs on the *y*-axis were transformed into the percentage in Equation (3). The residuals can be computed in Step A.For example, the θ (0 ≅ IP) on the scale (*x*-axis) can be obtained using Equation (3) when the day is 50 and the total infected days is 100. The OP_50_ = 0.5 if the CNICi is 50 cases and when Max is 100, Min is 0, and the footnote *i* in Equation (2) is 50 on the *x*-axis. The Epi in Equation (1) is determined by the known parameters a (=1) and b (=0) after performing Step E. The model residual is equal to 0 at the θ (=0) on the *x*-axis in this example.E.Perform the Solver add-in:The Microsoft Solver add-in [34,35,36] was used to estimate the model parameters (Appendix B). The OC can be plotted to predict the potential CNIC and to determine the IP days (see next section).

### 2.4. Searching IP

The IP search method is based on an IRT model fitted to the data. The IP is determined by the computation of the absolute advantage coefficient (AAC) or the dimension coefficient [37,38,39] in Equation (6).
(6)$$\mathrm{AAC}=\frac{\frac{\gamma_{3}}{\gamma_{2}}}{\frac{\gamma_{2}}{\gamma_{1}}},$$
where AAC is determined by the three consecutive EPi (denoted by $\gamma_{1},$
$\gamma_{2},\;\mathrm{and}\;\gamma_{3}\;\mathrm{in}\;\mathrm{Equation}\;\left(6\right);\;$Figure 1). The IP is then positioned at the minimum across all possible AACs on an OC.

### 2.5. Tasks to Validate and Present Data

#### 2.5.1. Task 1: Comparison of Two Scenarios in IRT Models

Not all cases involve the four phases of the OC during the COVID-19 pandemic (Figure 1). The best-fit model (with minimum total residual), for example, might be parameterized at Stage II (i.e., Epi = 0.5 = compressed coefficient [CR] at θ = 5) or earlier at Stage I. The OPi and the CNICi are, therefore, redefined by Equations (7) and (8).
(7)$$\mathrm{OPi}=\frac{\left(\mathrm{Oi}\;-\;\mathrm{Min}\right)}{\left(\mathrm{Max}\;-\;\mathrm{Min}\right)}\times {\mathrm{CR}}_{i},$$
(8)$$\mathrm{Expected}\;\mathrm{CNICi}=\mathrm{Epi}÷{\mathrm{CR}}_{i}\times \left(\mathrm{Max}\;-\;\mathrm{Min}\right)+\mathrm{Min}.\;$$

A paired *t*-test was performed to examine the differences in residuals between two scenarios (e.g., IRT and IRT–CR models).

#### 2.5.2. Task 2: The IPcase Index Used to Measure the SACOVID

We used the IPcase index (the square root of IP days multiplied by the corresponding CNIC) referring to the area (e.g., OcPd in Figure 1), a different approach to using the CNIC to measure the SACOVID. Only two values of IP day and CNIC were considered. The excessive days and CNICs were not included. This is similar to the bibliometric indicators (e.g., h- and x-index [5,6]) used to measure individual research achievements without considering the excessive sections of publications and citations. A choropleth map [40] was used to compare the ipcase-index for each country/region; a darker color represents more SACOVID on the basis of the IPcase index.

#### 2.5.3. Task 3: Comparisons of IP Days and CNICs in China

Choropleth maps were used to compare the IP days and CNICs of the provinces, metropolitan cities, and areas in China [40]. The Kano diagram [8,41] was used to describe the characteristics of the region toward CNIC-oriented, neutral, or IP-oriented.

The individual standard error (SE) determined by the root of the model residual was applied to estimate 95% confidence intervals using a forest plot [42,43]. The overall effect was determined by considering the weights of variance in individual areas. The Q-statistics and Z-score were used to examine the difference in IP days among areas in China.

The top three most-fit and misfit provinces, metropolitan cities, and areas were particularly selected to present their IP days on the OC plots.

### 2.6. Statistical Tools and Data Analysis

A visual representation of the forest plot displaying the comparison of the difference in IP days among provinces, metropolitan cities, and areas in China was plotted online on Google Maps. The IRT modeling process was executed in Microsoft Excel (Appendix B and Appendix C). The IRT mode for diagnosis in COVID-19 for regions was proposed.

## 3. Results

### 3.1. Comparison of Two Scenarios in IRT Models

The residuals are significantly different between the two study scenarios (t = 3.62, df = 32, *p* = 0.0005) with means (0.83 and 1.16) and variances (0.58 and 0.95) for IRT and IRT–CR models, repsectively, indicating that the CR plays a critical role in making the IRT–CR model possess fewer residuals than the two-parameter IRT model. The following analyses were based on the IRT–CR model.

### 3.2. Density of IPcase Index Around the World

Figure 2 shows the density of IPcase indices across the globe. We see that the top three are from India, Russia, and Brazil, with indices of 35,243, 32,415, and 31,897, respectively, connected by three blue lines. China was effected relatively mildly by COVID-19 compared with other countries/regions in the world on the basis of the colors shown in Figure 2.

### 3.3. The Most Infected Case Numbers and Longer IP Days in China

We infer that the majority of CNICs on the basis of the two perspectives of IP days and CNICs separated individually were in Hubei (including Wuhan), followed by Hong Kong and Guangdong (Figure 3). Shanghai has longer IP days, followed by Liaoning and Hong Kong (Figure 4). The NCIC in Figure 3 is not the corresponding NCIC on the basis of the IP days.

### 3.4. Association between IP and the Corresponding CNIC

All provinces/metropolitan cities/areas are distributed in a Kano diagram in Figure 5. We can see the IPs and their corresponding CNICs on the x- and y-axes. Hong Kong, Shanghai, and Hubei are the top three entities with the highest IPcase index sized by bubble at 1744, 723, and 698, respectively. Hubei (including Wuhan) has a lower IP day (18) but a large CNIC (27,100). The Kano diagram [8,41] could be complementary to the choropleth map [31], which cannot differentiate whether the entity attribute leans toward the IP in green, CNIC in red, or the neutral in yellow bubbles [7,8].

### 3.5. Using the Forest Plot to Compare IPs

A forest plot was used to compare the IPs sorted by the model residuals (Figure 6). A distinct difference in IP days was determined (χ2 = 5,065,666, df = 32, *p* < 0.001; Z ≤ 100, *p* < 0.001) among areas in China.

We observed that the top three areas having aberrant CNIC patterns were Yunnan, Sichuan, and Tianjin, with IP days at 5, 51, and 119, respectively (Figure 7). The provinces of Anhui, Hunan, and Henan had the most model-data-fit regions (having a smaller number of residuals) with IP days of 12, 11, and 12, respectively (Figure 8). The IP-day variable for Hubei (including Wuhan) was 18.

### 3.6. Online Dashboards Shown on Google Maps

All those line plots would appear once the area in the choropleth maps (e.g., Figure 2, Figure 3 and Figure 4) are clicked using the links in [36,37,38]. Similarly, those links [44,45,46,47,48] regarding the plots in Figure 5 and Figure 6 are shown in Google Maps.

## 4. Discussion

### 4.1. Principal Findings

A difference was found in residuals between IRT–CR and IRT models, indicating that the IRT–CR model better fit the epidemic data compared to the IRT model. More parameters make the model a better fit for the data [49,50].

We used the IPcase index to analyze the impact of COVID-19 (Figure 5). The results showed that Hong Kong, Shanghai, and Hubei are the top three cities most affected by COVID-19, with IPcase indices of 1744, 723, and 698; respectively, quite different from the results in IP or CNIC (Figure 3 and Figure 4) alone in analyses. Using the single CNIC [2,3] to compare the negative impact of COVID-19 is problematic and unreasonable.

The provinces of Yunnan, Sichuan, and Tianjin are the top three entities with the most aberrant patterns, with IP days of 5, 51, and 119, respectively, suggesting that the irregular response patterns are worthy of further investigation using person-fit statistics in reporting and analyzing data [51,52,53]. The most model-data-fit areas were Anhui, Hunan, and Henan, with IP days of 12, 11, and 12, respectively, including Hubei (and Wuhan) with 18 IP days, similar to South Korea, which has successfully maintained a flat infection curve [13]. However, the IP-determination scheme is scientifically and appropriately suitable as a test of effective control of COVID-19, compared to the eyeball method used in a previous study [13].

Google Maps show and demonstrate an online dashboard that compares the IP days and IPcase index across provinces, metropolitan cities, and areas in China [44,45,46,47,48]. This is a modern and innovative data representation and is better than the traditional static-image display [54].

### 4.2. Contribution of This Study

No studies have used both IP days and the corresponding CNIC (or IPcase index) to analyze the effective control of COVID-19, aside from the CNIC and the fatality rate that were used to evaluate the SACOVID [2]. We used the principle of bibliometric indicators (e.g., h- and x-index [5,6]), considering both IP and CNIC dimensions for the evaluation of SACOVID. Figure 2 shows the IPcase index on the choropleth map, and the OC appears when the region of interest is clicked. The IPcase index can be applied to the interactive web-based dashboard [55] (developed by Johns Hopkins University to track COVID-19 in real-time) and other epidemic fields in the future.

Several mathematical models [16,17,18,19,20,21,22,23] have been proposed to predict the CNIC, and some IP determinations have been addressed [9,10,11,12,13], but none used the IRT to develop a predictive model during the COVID-19 pandemic using the IP search and compare the IPs (or propose the IPcase index) in practice. The epidemic CNIC in most regions in this study can be fitted rather well using the IRT–CR model. The IP search method [37,38,39] is unique and feasible compared with methods using the average of the previous several daily numbers of confirmed cases (e.g., the mean in recent 2 or 7 days) [12,25].

The choropleth map [39] can be complemented by a Kano diagram [8,41] with more IP or CNIC attribute details. The online forest plot [42,43] is another feature that compares the SACOVID across regions. Google Maps show an online dashboard that compares the impact hit of COVID-19 across the entities [36,37,38]. Readers are invited to examine them in detail on their own dashboards.

### 4.3. What It Implies and What Should Be Changed

Over 96,500 articles published in the PubMed database were searched by using the keyword “COVID-19” in the title [56]. No such comparisons were made using the IPcase index (or IP days) to present the impact hit by COVID-19, until now.

The uses of novel graph-based data models [57] and recommendation techniques [58] have shown promise in recent years. The online-dashboard-type representation used in epidemiology is proposed for future studies, and not limited to the COVID-19 pandemic as we performed in this study.

The capacity for effective control of COVID-19 should be calculated as the one [13] reported by South Korea, which has successfully maintained a flat infection curve for more than 50 days. However, if South Korea is selected on the choropleth map [44] and the newly computed IP is applied up to 23 November 2020, the IP days is 345, which is considerably longer than the IP in Hubei (China) at 18. The reason is the second wave of COVID-19 occurring in South Korea at the end of 2020.

The animated dashboards designed for this study also surpass the static images in relevant articles [54]. One picture is worth 1000 words [59]. We hope that future related research will be able to make use of Kano diagram visualizations, choropleth maps, and forest plots displayed on dashboards as we have in this study.

### 4.4. Strengths of This Study

First, the comparison of effective control of COVID-19 in regions can be calculated using the IPcase index in our proposed IP scheme using AAC to search for IP on a given OC. Aberrant misfit model areas (e.g., those in Figure 7) should be further explored because their model residuals deviated significantly from our expectations.

Second, no MP4 videos on how to model CNIC and estimate the parameters have been given to readers for replicating the study in the future, particularly MS Excel readers.

Third, using the Microsoft Solver add-in is a common approach, but only a few were shown in previous studies [34,35,36] to model the epidemic situation for each region. Data and model building videos in Microsoft Excel are provided in Appendix A and Appendix C. The approach of searching IP days for regions affected by COVID-19 is easy to understand.

Fourth, the IRT model applied to diagnose the most misfit regions in COVID-19 for policymakers is the merit of this study.

### 4.5. Limitations and Future Studies

Our study has some limitations. First, data were downloaded from Google Sheets on a daily basis. Many regions with mild and asymptomatic cases were not detected and documented [60,61,62,63,64]. For instance, SARS-CoV-2 may exist in a population without clinical cases for a long period [64]. The model building and IP search would be biased.

Second, the minimal AAC [37,38,39] is defined as the location of IP on an OC that will affect the corresponding CNIC to compute the IPcase index. Although the AAC is objective and viable, it is necessary to compare the difference in effect using the mean of the previous several daily case numbers (e.g., within 2 or 7 days) [12,24] in the future.

Third, the case number is changeable and varied day by day. The model parameters during the COVID-19 pandemic in countries/regions should be optimized on a daily or weekly basis to make the ipcase-index as accurate as possible.

Fourth, the Microsoft Solver add-in is not a unique approach to estimate model parameters. Many other methods can be applied to estimation, such as warm’s weighted mean likelihood estimate [65], anchored maximum likelihood estimation [66], and weighted likelihood estimation [67]. They are worthy of comparison in the future.

Fifth, visual dashboards are shown on Google Maps. However, these achievements are not free of charge. For example, the Google Maps application programming interface (API) requires a paid project key for the cloud platform. Thus, the limitations of the dashboard are that it is not publicly accessible and it is difficult to mimic by other authors or programmers for use in a short period of time.

Last, although IRT is common and popular in the educational and psychometric field, many readers in public health are unfamiliar with IRT. The IRT–CR model consists of two parameters and CR adjustments in the model build that need some effort to understand and mimic through data and MP4 videos provided in Appendices.

## 5. Conclusions

We used the IRT model to assess the IPcase index for each region on the negative impact of COVID-19, and to compare the difference in SACOVID between all regions of the world and China. Three visual representations of the choropleth map, the Kano diagram, and the forest plot were demonstrated to display the results for a better understanding of the comparison of disease outbreak situations, including the diagnosis of best-fit and misfit regions during the COVID-19 epidemic. The IRT model incorporated with the AAC is recommended for other epidemic outbreaks for determining the IP and the corresponding CNIC, not just limited to the COVID-19 as illustrated in this study.

## Figures and Tables

**Figure 1 ijerph-18-01994-f001:**
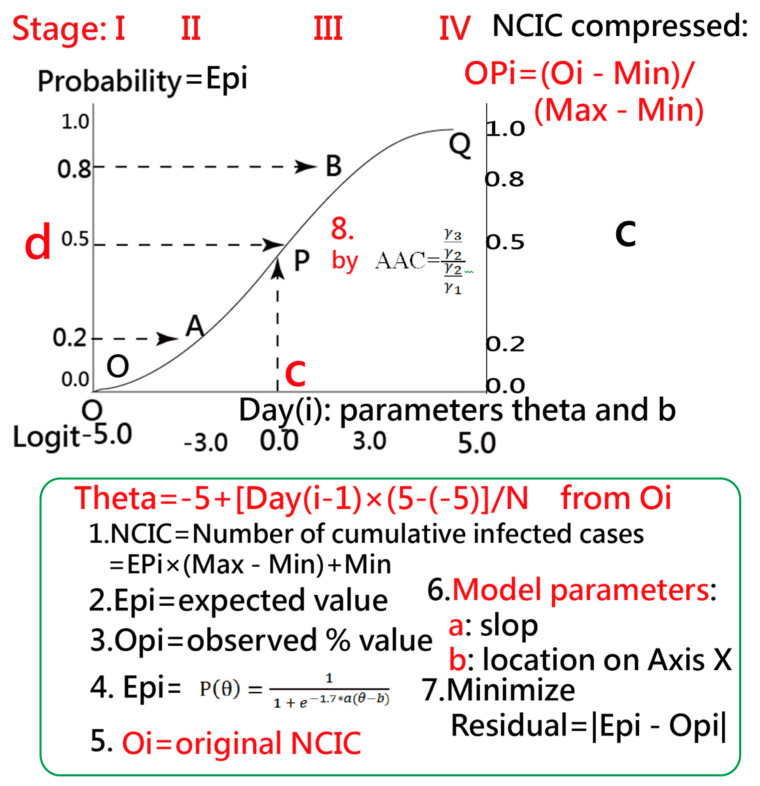
Determining the Inflection Point (IP) on an ogive curve and the ipcase-index denoted by the four-point OcPd rectangle using the Item Response Theory model.

**Figure 2 ijerph-18-01994-f002:**
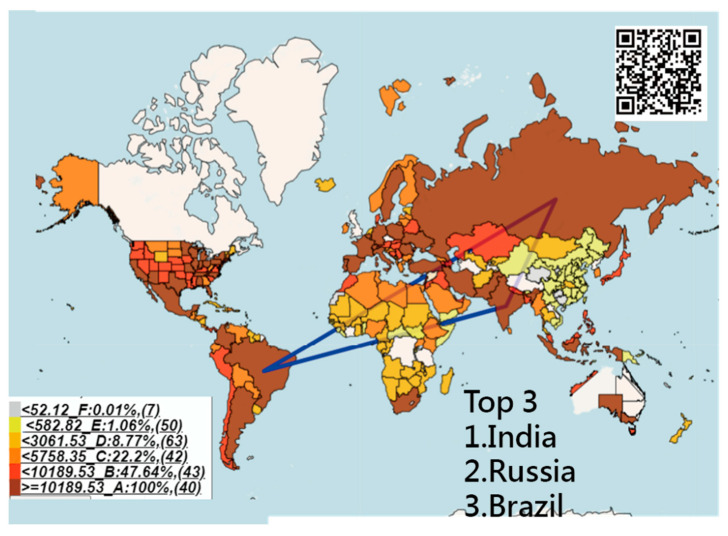
The density of the IPcase index used to measure the SACOVID on countries/regions in the world.

**Figure 3 ijerph-18-01994-f003:**
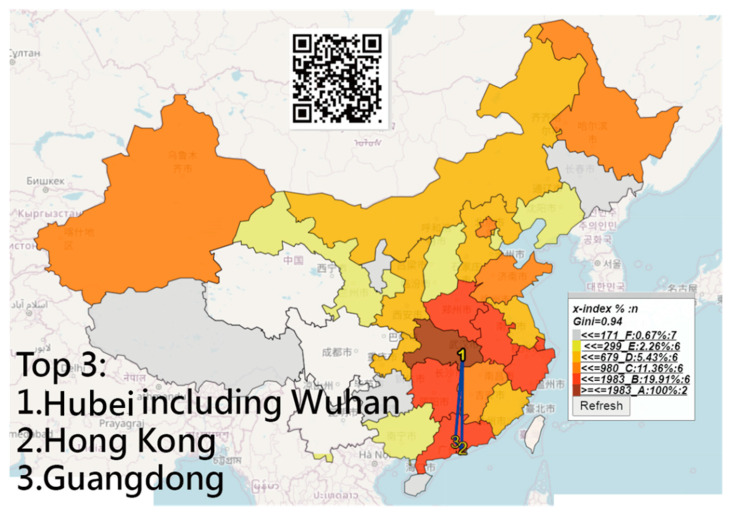
The top three regions with the most CNICs in China.

**Figure 4 ijerph-18-01994-f004:**
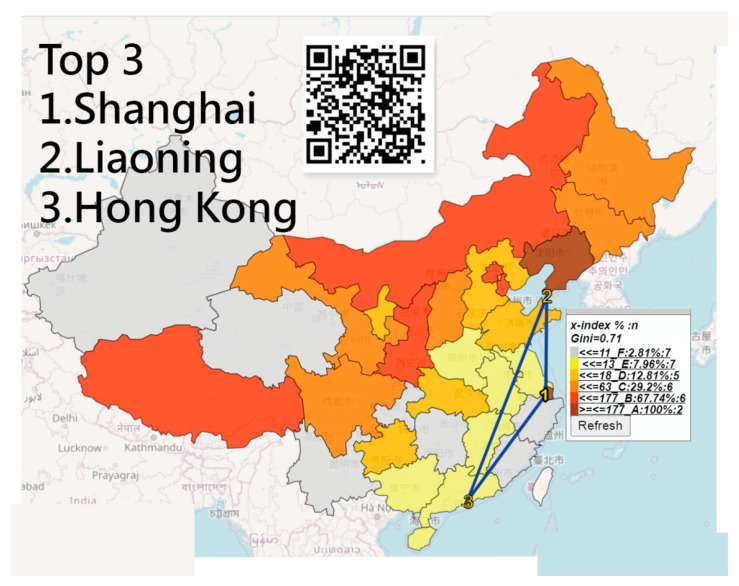
The top three regions with the most inflection point (IP) days in China.

**Figure 5 ijerph-18-01994-f005:**
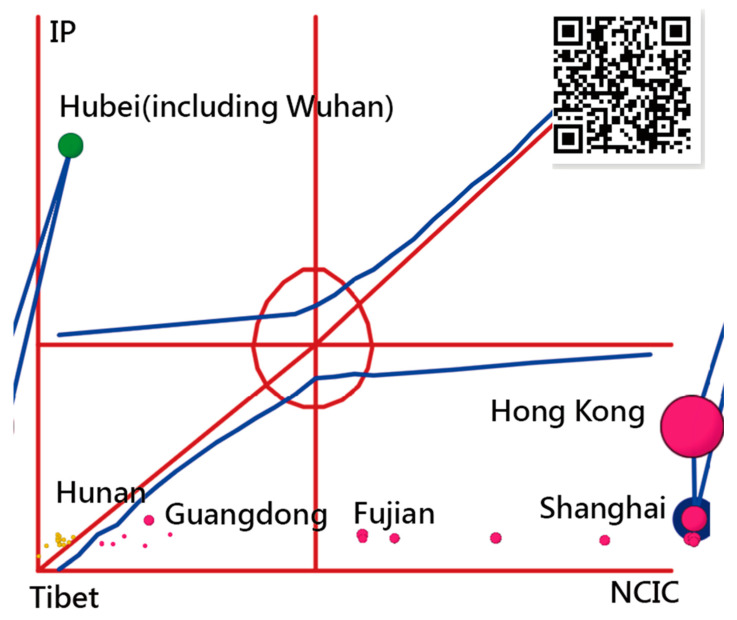
The attribute of ipcase-index toward IP or CNICs against SACOVID in China shown in a Kano diagram.

**Figure 6 ijerph-18-01994-f006:**
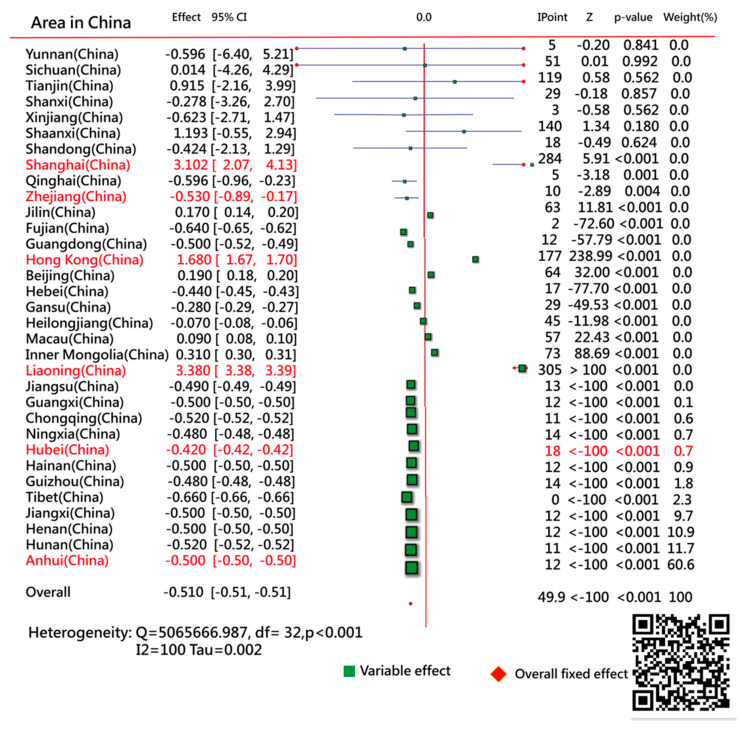
Comparison of IP days among regions in China using the forest plot.

**Figure 7 ijerph-18-01994-f007:**
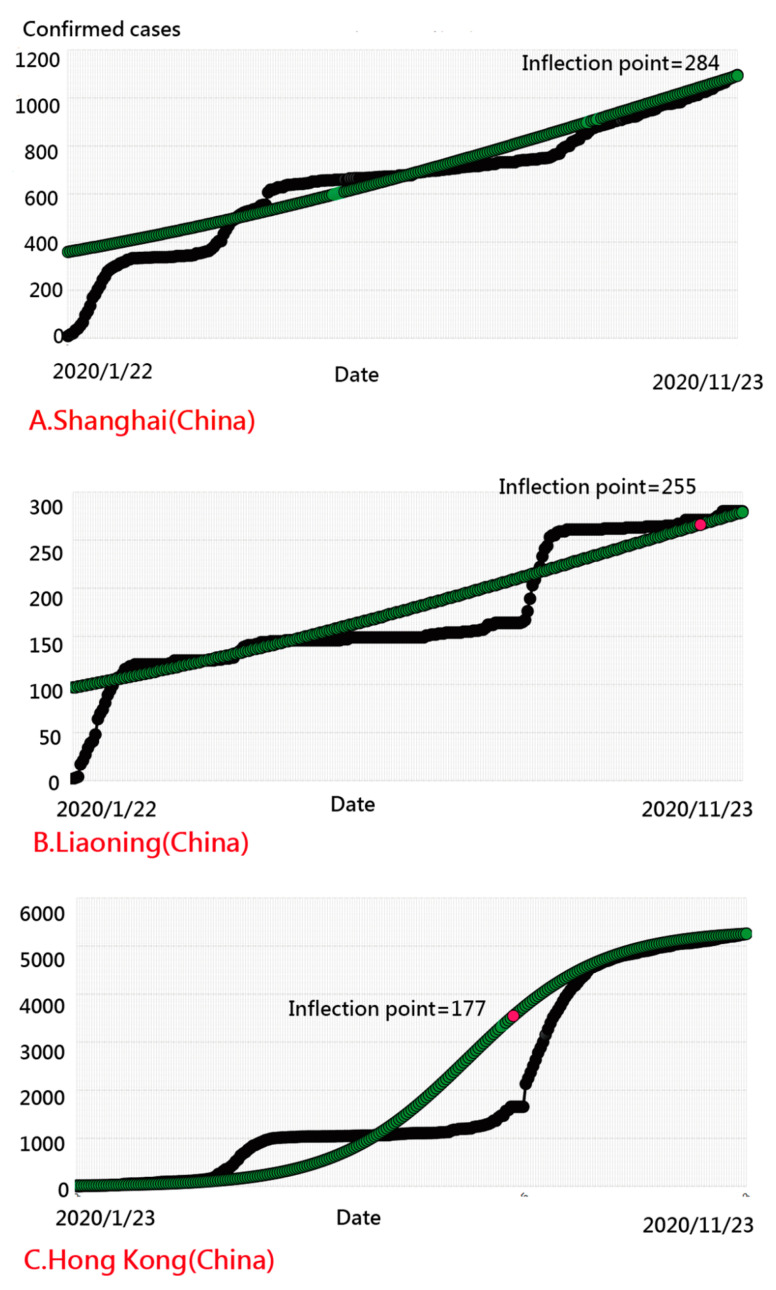
Regions with the most inaccurate curve on COVID-19 in China.

**Figure 8 ijerph-18-01994-f008:**
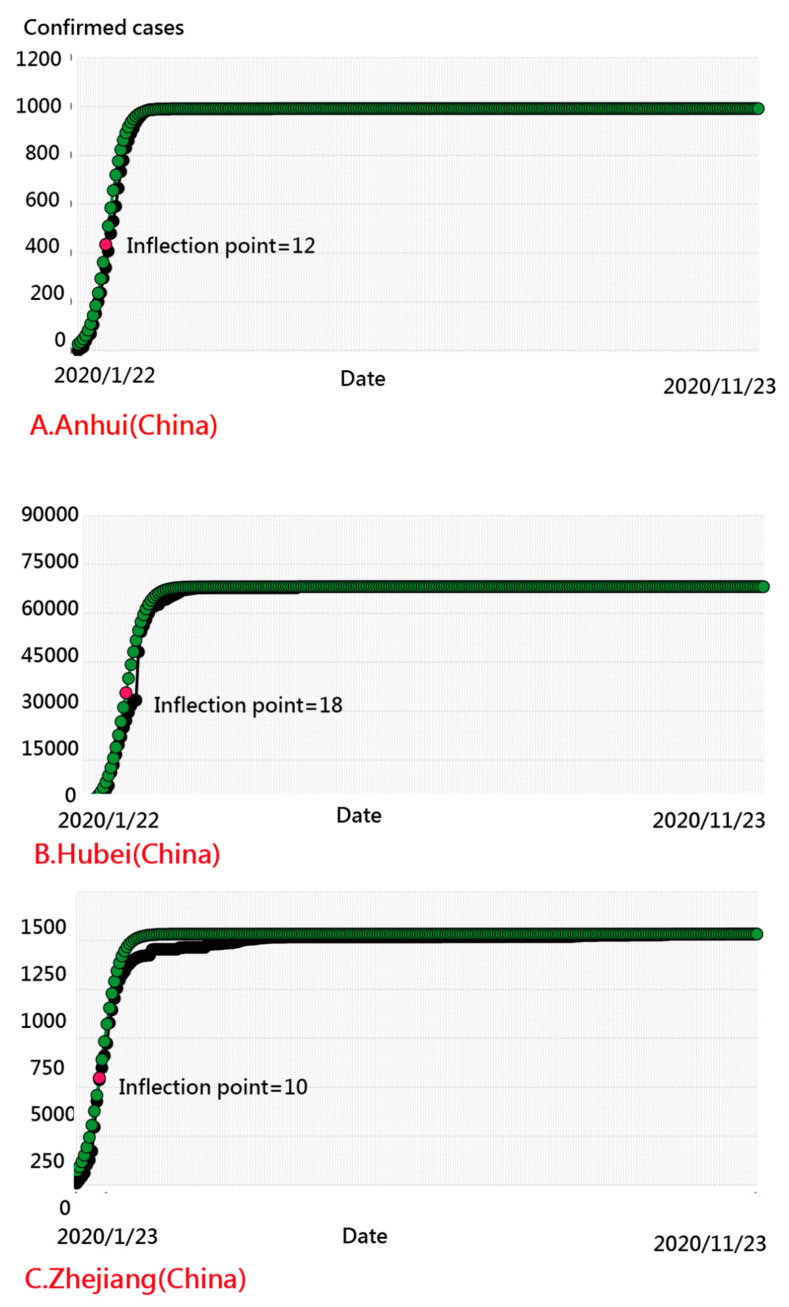
Regions with the best-fit curve on COVID-19 in China.

## Data Availability

All data were deposited in Appendices.

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
