# Peer review of "Using the IPcase Index with Inflection Points and the Corresponding Case Numbers to Identify the Impact Hit by COVID-19 in China: An Observation Study"

_ijerph, 2021, doi:10.3390/ijerph18041994_

Round 1

Reviewer 1 Report

The authors investigate the impact hit by COVID-19 in China through the ipcase-index with inflection points and the corresponding case numbers.

The proposed study is interesting but there are some points that the authors should better discuss.

The authors should be better described the novelties of their study with respect to existing ones. In particular, the author should discuss limitation and cons of the examined approaches. Furthermore, the authors should provide more details and discussion about the obtained results. The Discussion section also needs to be improved by analyzing the outcome of evaluation section.

I suggest to further analyze more recent approaches about the examined topics. In particular, I suggest the following papers to further investigate the graph-based machine learning and emotional features during the COVID pandemia:

1) An emotional recommender system for music. IEEE Intelligent Systems.

2) DICO: a graph-db framework for community detection on big scholarly data. IEEE Transactions on Emerging Topics in Computing.

Finally, I suggest to perform a linguistic revision.

Author Response

Reviewer 1:

The authors investigate the impact hit by COVID-19 in China through the ipcase-index with inflection points and the corresponding case numbers.

The proposed study is interesting but there are some points that the authors should better discuss.

The authors should be better described the novelties of their study with respect to existing ones. In particular, the author should discuss limitation and cons of the examined approaches. Furthermore, the authors should provide more details and discussion about the obtained results. The Discussion section also needs to be improved by analyzing the outcome of evaluation section.

Response:

  • In Discussions 4.4, we illustrated several points regarding the novelties in this study.
  • In Limitations, we have raised several points that should be cautious in future studies.
  • As suggested by the reviewer, we have added several sentences to elucidate the obtained results in this study.
  • Additional limitation was added to this manuscript. That is, visual dashboards are shown on Google Maps. However, these achievements are not free of charge. For example, the Google Maps application-programming-interface(API) requires a paid project key for use on the cloud platform. Thus, the limitation of the dashboard is that it is not publicly accessible and is difficult to mimic by other authors or programmers for use in a short period of time.

I suggest to further analyze more recent approaches about the examined topics. In particular, I suggest the following papers to further investigate the graph-based machine learning and emotional features during the COVID pandemia:

1) An emotional recommender system for music. IEEE Intelligent Systems.

2) DICO: a graph-db framework for community detection on big scholarly data. IEEE Transactions on Emerging Topics in Computing.

Response: We have added the recommended articles into references and Discussions for enriching the content and implication in this revised manuscript.  

Finally, I suggest to perform a linguistic revision.

Response: English in this manuscript has been revised by a native English speaker.

Reviewer 2 Report

The article titled "Using the ipcase-index with Inflection Points and the Corresponding Case Numbers to Identify the Impact Hit by COVID-19 in China: Observation study" has the goal to

(a) develop an index that can be used to evaluate the impact of COVID-19,

(b) apply item response theory model to determine the Inflection point days, and

(c) finally compare the difference in the impact of SACOVID (struggle to fight against COVID-19) across all areas in China.

The goals of this paper are interesting and provides overall insight as the most hit areas being Hong Kong, Shanghai, and Hubei, and results could have solid impact to COVID-19 policy making, but there are a few things that need to be addressed. The language of the manuscript is difficult in some places, convoluted sentences, with a consistent issue with spacing, spelling and use of acronyms. In addition, the references have not consistent outlay. This manuscript would greatly improve with a native English speaker's revision. For example, is it supposed to be CNIC or NCIC? 

In regards to methods, the overall development of the index needs to be clearly laid out for one to follow it and more details are needed for each figure; for example, Figure 1 and others would benefit if a more descriptive narrative is taken. An example of the model by providing an example would benefit the understanding applying IRT to determine the IP. The video https://www.youtube.com/watch?v=Xj9pJMxfs0o has no sound to it so it is difficult to follow step by step.

In addition, the figures in the manuscript as well as some text are pixelated, which could be easily improved. 

As for the discussion, there is too much repeating of the results and introduction, and no true discussion of the results. For example, what is the impact of findings and as compared with others work. Authors suggested in the abstract their application of IRT model with AAC to determine IP days is not limited to COVID-19, but there is not discussion on this. 

With more descriptions and overall editing of the manuscript, the full impact of the manuscript can be appreciated. 

Author Response

Reviewer 2:

The article titled "Using the ipcase-index with Inflection Points and the Corresponding Case Numbers to Identify the Impact Hit by COVID-19 in China: Observation study" has the goal to

(a) develop an index that can be used to evaluate the impact of COVID-19,

(b) apply item response theory model to determine the Inflection point days, and

(c) finally compare the difference in the impact of SACOVID (struggle to fight against COVID-19) across all areas in China.

The goals of this paper are interesting and provides overall insight as the most hit areas being Hong Kong, Shanghai, and Hubei, and results could have solid impact to COVID-19 policy making, but there are a few things that need to be addressed. The language of the manuscript is difficult in some places, convoluted sentences, with a consistent issue with spacing, spelling and use of acronyms. In addition, the references have not consistent outlay. This manuscript would greatly improve with a native English speaker's revision. For example, is it supposed to be CNIC or NCIC?

Response: English in this manuscript has been revised by a native English speaker and the references have been rechecked and matched to the inner context in the revised manuscript.   

In regards to methods, the overall development of the index needs to be clearly laid out for one to follow it and more details are needed for each figure; for example, Figure 1 and others would benefit if a more descriptive narrative is taken. An example of the model by providing an example would benefit the understanding applying IRT to determine the IP. The video https://www.youtube.com/watch?v=Xj9pJMxfs0o has no sound to it so it is difficult to follow step by step.

Response: As suggested by the reviewer, we have overly rewritten and reorganized the section of Methods for making the approach of modelling parameters as clear as possible. The Additional files with the original Excel model and MP4 videos that could be helpful for readers who are interesting in replicate their studies on their own in the future.

  We divided the Methods into three parts: (1) IRT concept, (2) model building and parameter estimation, and (3) data representations matched with the results in Figures. Hopefully, the organized structure in Methods that can be readable and comprehensive more than the original version of manuscript.

   Due to the language barrier, the MP4 videos with no sound. Nonetheless, these three Appendices are enough to make the model building and parameter estimation clear and understandable to replicate another similar studies for interested readers.    

In addition, the figures in the manuscript as well as some text are pixelated, which could be easily improved.

Response: QR-codes in Figures are provided to readers who can scan them and examine the detail in Figures. In addition, all those Figures have been reproduced to upgrade the resolution in Figures.

As for the discussion, there is too much repeating of the results and introduction, and no true discussion of the results. For example, what is the impact of findings and as compared with others work. Authors suggested in the abstract their application of IRT model with AAC to determine IP days is not limited to COVID-19, but there is not discussion on this.

Response: The main purpose is to illustrate the IP determination and the model parameters estimation and How to application in COVID-19 is exampled in China. We have made improvement in Discussions according to the recommends from the reviewer regarding some resulting findings in Figures discussed in this revised manuscript.

With more descriptions and overall editing of the manuscript, the full impact of the manuscript can be appreciated.

Response: We have made a lot of revisions in the revised manuscript. Hopefully, the revised version can meet and reach the reviewer’s concerns.

Reviewer 3 Report

The authors propose an index to evaluate the impact of COVID-19 in different countries and Chinese provinces. he method proposed seems interesting. However, the paper is poorly written and organized. Hence, I believe that a major revision is needed, before it is possible to go in to the details of its contribution. In particular, I have the following major comments:

  • In general, I suggest the authors to carefully revise the paper, since there are several paragraphs that are hard to follow, and the narrative has not a good flow.
  • In the Introduction, the authors discuss the limits of the methodologies currently used in the literature. However, they do not explain what is the contribution of this paper. i believe that a good introduction should present the main ideas and finding of the paper
  • The Materials and Methods are poorly organized and hard-to-read. There are formulas introduced without any clear motivation and explanations
  • The results seem interesting. However, a proper discussion on their meaning seems to miss, according to this reviewer
  • The conclusions should highlight the main take home message of the work. To me, this is not done in the current version

Author Response

Reviewer 3:

The authors propose an index to evaluate the impact of COVID-19 in different countries and Chinese provinces. The method proposed seems interesting. However, the paper is poorly written and organized. Hence, I believe that a major revision is needed, before it is possible to go in to the details of its contribution. In particular, I have the following major comments:

In general, I suggest the authors to carefully revise the paper, since there are several paragraphs that are hard to follow, and the narrative has not a good flow.

In the Introduction, the authors discuss the limits of the methodologies currently used in the literature. However, they do not explain what is the contribution of this paper. i believe that a good introduction should present the main ideas and finding of the paper

Response: (1) In the revised Introduction, we explained the challenge in methodologies for determining IP at the beginning and the goal of this study later on.

         (2) The main ideas were illustrated (e.g., the aims of this study).

         (3) The finding of this study could not be explained in Introduction due to the work has not been proceeded.

The Materials and Methods are poorly organized and hard-to-read. There are formulas introduced without any clear motivation and explanations

Response: As suggested by the reviewer, we have rewritten and reorganized the Methods to make the approach easily to follow and understand.

    We divided the Methods into three parts: (1) IRT concept, (2) model building and parameter estimation, and (3) data representations matched with the results in Figures. Hopefully, the organized structure in Methods that can be readable and comprehensive more than the original version of manuscript.   

The results seem interesting. However, a proper discussion on their meaning seems to miss, according to this reviewer

Response: We have also rewritten the section of Discussions. Hopefully, the revised version can meet the reviewer’s concerns.

  In addition, the study focused on two aspects of (1) how to estimate model parameters and (2) how to determine the IP days. The ipcase-index is proposed to describe the drawback of usage in COVID-19 just with one dimension of CNIC or IP days(e.g., flattening the ogive curve). The demonstrations of regions in China using visual representations is trivial and not vital as the methodology introduced in this study. As such, the discussion on outcome in Figures is toward methodology instead of epidemic in China.      

The conclusions should highlight the main take home message of the work. To me, this is not done in the current version

Response: As suggested by the reviewer, the conclusion has be rewritten to make the manuscript more clear and easy to highlight the spot and contribution of this study to readers. 

Round 2

Reviewer 2 Report

The second version was vastly improved to demonstrate clarity with the methods section and the methodology used in the study. The improvements in the language is seen, but minor editing with spacing is still needed. Find attached yellow highlighted areas that need spacing considerations. The QR-codes added to the figures could resolve the issue of pixelated images online, but does not help with printed copies. Lastly, a consistent format for references is still needed. There are references that show doi's, and PMIDs or PMCID. but not in all references.

Author Response

Response: Spaces have been revised in the revised manuscript. The QR-codes provided in the manuscript that can help the detail online, not limited to the printed copies. Usually, the doi;s and PMIDs or PMCID are not necessary in references due to other information of Title and authors being retrieved in academic databases.

Reviewer 3 Report

I am mostly satisfied by the author's revision.

Author Response

Response: The revised manuscript with additional improvement was submitted to the IJERPH.